# Cassini: streamlined and scalable method for in situ profiling of RNA and protein

Nicolas Lapique[1,5] ✉, Michael Taewoo Kim[1,2,3,5], Nicholas Thom[1,2,3], Naeem M. Nadaf [1], Juan Pineda[1] & Evan Z. Macosko [1,4] ✉

In the expanding field of spatial genomics, numerous methods have emerged to decode biomolecules in intact tissue sections. Advanced techniques based on combinatorial decoding can resolve thousands of features in a reasonable time but are often constrained by either the prohibitive costs associated with commercial platforms or the complexity of developing custom instruments. Alternatively, sequential detection methods, like single-molecule FISH, are easier to implement but offer limited multiplexing capability or signal amplification. Here, we introduce Cassini, an innovative approach for straightforward, cost-effective multiplexed measurements of mRNA and protein features simultaneously. Cassini leverages rolling circle amplification, known for its robust amplification and remarkable stability even after intense stripping, to serially detect each feature in under 20 minutes. The method also enables simultaneous immunostaining with either fluorophore-conjugated or DNA-barcoded antibodies, through an optimized immunostaining buffer. In a single overnight run, we show that Cassini can quantify 32 features (comprising both RNA and proteins) with sensitivity similar to state-of-the-art FISH techniques. We provide a comprehensive protocol alongside an online probe-design platform (cassini.me), aiming to enhance accessibility and user-friendliness. With our open-source solution, we aspire to empower researchers to uncover the nuances of spatial gene expression dynamics across diverse biological landscapes.

Imaging-based spatial genomics technologies enable surveys of biomolecular characteristics in intact tissues at submicron resolution[1,2]. There are primarily two types of methods. One involves combinatorial detection, where each round of detection exponentially increases the number of detectable features[3–9]. In this detection approach, thousands of transcripts can be examined in fewer than 20 rounds. The other strategy utilizes sequential detection, resulting in a linear increase in the number of identified features with each hybridization round[10–13], consequently exhibiting much lower multiplexing capacity compared to combinatorial methods. Nevertheless, sequential methods remain prevalent due to their cost-effectiveness, ease of

implementation, and ability to achieve the highest sensitivity. State-of-the-art sequential methods employ signal amplification to improve the signal-to-noise ratio[10–12]. However, each round typically necessitates the re-amplification of the endogenous target molecules, a process that can exceed 24 h of experimental time[10] or requires complex temperature control setups[11]. Other methods, such as osmFISH[13], have a more straightforward cycling regime but suffer from low signal amplification, compromising the signal-to-noise ratio.

A major opportunity associated with spatial genomics platforms is the ability to simultaneously profile gene expression and protein expression in the same assay. Existing microscopy based methods for

[1]Broad Institute of Harvard and MIT, Cambridge, MA, USA. [2]Graduate School of Arts and Sciences, Harvard University, Cambridge, MA, USA. [3]Division of Medical Science, Harvard Medical School, Boston, MA, USA. [4]Department of Psychiatry, Massachusetts General Hospital, Boston, MA, USA. [5]These authors contributed equally: Nicolas Lapique, Michael Taewoo Kim. ✉e-mail: nlapique@broadinstitute.org; emacosko@broadinstitute.org

accomplishing this suffer relying on conventional ISH probe with limited amplification[6] or low multiplexing capacity[12,14]. One technical challenge has been the difficulty with combining enzymatic amplification strategies like rolling circle amplification (RCA) with immunostaining that relies on buffer ingredients that can inhibit downstream enzymatic activity. For example, dextran sulfate is used to enhance the staining specificity of DNA-conjugated antibodies[15,16], but this potent enzymatic inhibitor[17] renders downstream enzymatic amplification very challenging. Although some methods have demonstrated simultaneous detection of RNA and protein proximity using enzymatic amplification[18,19] these are not applicable to conventional single-protein immunostaining.

Here, we present a method for spatial profiling of RNA and protein, which we call Cassini. Our system combines strong signal amplification that can withstand harsh probe stripping, with fast cycling times for a simpler, more straightforward cycling approach. Cassini relies on the ability of PBCV-1 DNA ligase (also known as SplintR ligase) to directly ligate Padlock probes onto RNA templates[20,21]. After probe amplification through rolling circle amplification (RCA), the individual gene sequences are directly detected using a specific fluorescent probe. Previous work demonstrated that RCA using SplintR ligase enables multiplexed mRNA detection[22], however this method is limited to RNA detection. To enable truly multimodal and multiplexed analysis using padlock probes, we devised: (1) a staining buffer to mitigate enzymatic inhibition while preserving specificity; and (2) a post-staining fixation protocol to ensure the retention of the signal. To our knowledge, this is the first report of a custom immunostaining buffer that preserves the specific binding of oligo-conjugated antibodies while maintaining compatibility with enzymatic activity. Cassini can detect simultaneously mRNA, conjugated antibodies, and conventional immunostaining and achieves precise spatial mapping of transcripts and proteins. Following probe amplification, we demonstrate the ability to map an entire adult mouse brain hemisphere at a rate of 18 min per gene at a low per-sample cost. By leveraging accessible technologies, openly shared protocols and providing a web interface to design the probes (cassini.me), we circumvent the barriers of instrumentation and cost.

## Results

It has been previously shown that padlock probes can be efficiently ligated and amplified on mRNA template using SplintR ligase[20–22]. First, we confirmed that RCA products remained fixed in situ after multiple rounds of stripping with 80% formamide (Supplementary Fig. 1a) with the estimated displacement of the foci being below 200 nm (Supplementary Fig. 1b). Additionally, we confirmed that the foci count remains constant throughout the cycles, demonstrating that the sensitivity remains stable (Supplementary Fig. 1c). However, we discovered that the method strongly compromised conventional immunostaining, suggesting that the antibody is being washed away during the process (see green bars in Fig. 1b and Supplementary Fig. 2a). We hypothesized that fixing the antibodies after immunostaining would prevent them from being dislodged (Fig. 1a top). We therefore post-fixed the antibodies with 4% paraformaldehyde for 2 h; this modification resulted in the complete restoration of immunostaining signals (orange bars in Fig. 1b and Supplementary Fig. 2a), regardless of whether the initial staining was strong (Fig. 1b left), moderate (Fig. 1b center), or weak (Fig. 1b right). We also confirmed that post-fixation does not alter the expected spatial distribution of the immunostaining signal (Supplementary Fig. 2a). The quality of the immunostaining remains consistent with the conventional method (Supplementary Fig. 2b, left), whether the secondary immunostaining is performed prior to the second fixation and amplification (Supplementary Fig. 2b, center) or after the entire process is completed (Supplementary Fig. 2b, right). We next tested DNA-conjugated antibodies with our updated fixation protocol; similar to the conventional

immunostaining, the signal was retained post fixation (Supplementary Fig. 3).

The DNA-conjugated antibodies frequently employed to increase multiplexing capabilities have an additional challenge associated with their use. Blocking buffers employed in conventional immunostaining do not provide sufficient specificity when used with conjugated antibodies (Fig. 1c left image and blue line in left panel). Typically, blocking buffer for conjugated antibodies employs high-molecular-weight (>500 K) dextran sulfate polymers to prevent off-target binding[15,16], but these polymers are highly inhibitory to most enzymatic reactions[17] including RCA with Phi29 (Fig. 1c center left image and red line in left panel). We tested a variety of alternative blocking additives and discovered that low-molecular-weight (~4 K) dextran sulfate can be used as a highly efficient blocking buffer for conjugated antibodies without interfering with the enzymatic amplification and displays very similar profile to conventional immunostaining (Fig. 1c right images and green and purple lines in left panel). Existing methods have employed RCA for detecting antibody–DNA conjugates either in vitro[23] or in proximity ligation assays[18,19] typically using similar buffers containing salmon sperm DNA and protein blockers. However, both the commercial buffer (Duolink) and those reported in the literature[18] exhibit substantially lower specificity compared to the Cassini buffer (Supplementary Fig. 4).

Due to the RCA amplification, the positive signal would be expected to be orders of magnitude higher when compared to staining with secondary antibodies. To match the optimal signal, the concentration of conjugated antibodies was decreased 100-fold compared to conventional immunostaining. Dual staining of the same antibodies with and without conjugation showed excellent overlapping signal (Fig. 1d) and correlations (Supplementary Fig. 5), with some key, interesting differences. Specifically, the conjugated antibodies produced a more punctate signal compared to immunostaining with conventional anti-NeuN antibodies, likely reflecting the localized signal amplified through rolonies (Fig. 1d left). Furthermore, conjugated antibodies showed less positive signal in thin subcellular structures like the astrocytic processes labeled by GFAP immunostaining (Fig. 1d right). The speckled background observed in the fiber tract appears in both conventional and conjugated NeuN immunostaining (Fig. 1d left and Supplementary Fig. 6), suggesting it is not specific to the conjugated method but likely reflects non-specific binding inherent to the antibody. To our knowledge, this is the first report of a custom immunostaining buffer that preserves the specific binding of conjugated antibodies while maintaining compatibility with enzymatic activity.

We next evaluated our method by comparing it to hybridization chain reaction FISH (HCR-FISH), a well-established reference technique for in situ hybridization[24,25]. Cassini produced larger foci (mean of $0.99 \pm 0.51\ \mu m^2$) than HCR-FISH (mean of $0.55 \pm 0.15\ \mu m^2$) (Fig. 1e and Supplementary Fig. 7), which can be advantageous in tissues with high autofluorescence background but could impair sensitivity in cases of overcrowding of highly expressed RNAs. We chose 7 genes with different levels of abundance and compared the measured density in selected areas (Fig. 1f, g and Supplementary Fig. 8), which was calculated by computing the local foci density within a $10\ \mu m$ radius around each spot. This quantification approach is advantageous compared to the total count or area density, as it is less sensitive to how the area is selected in each sample, tissue damage or imaging artifact. The foci count was performed by computing the local maxima, with the prominence adjusted to the spot intensity (Supplementary Fig. 9). Our method produced results that closely matched those of HCR-FISH, with only 2 out of 7 genes showing a significant $p$ value for local density differences (<0.05) (Fig. 1f and Supplementary Fig. 8), suggesting that the performance of Cassini is comparable regardless of expression density. To confirm this, we compared the expression gradient of *Amigo2* across the CA2-CA3

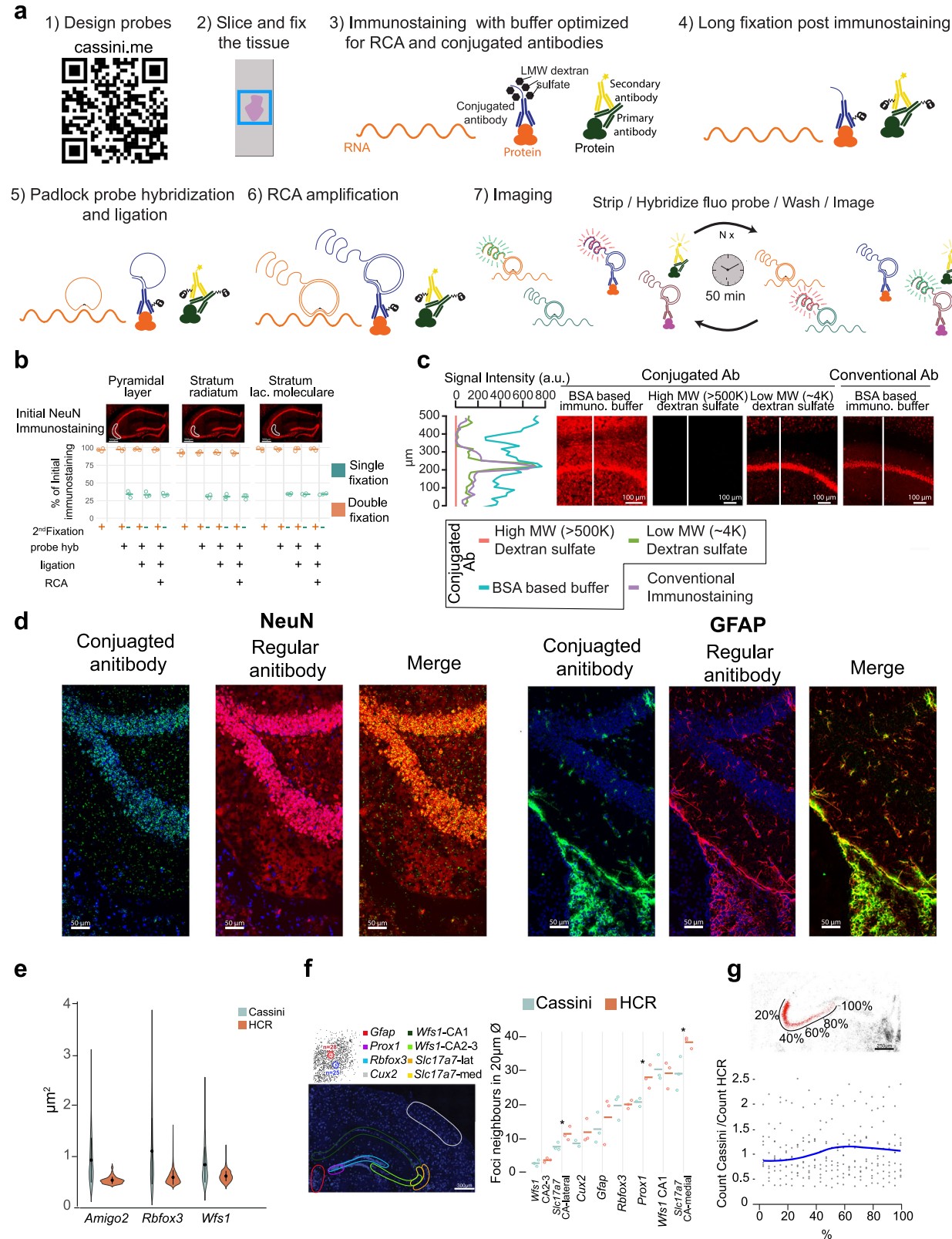

**a**
1) Design probes
cassini.me
2) Slice and fix the tissue
3) Immunostaining with buffer optimized for RCA and conjugated antibodies
4) Long fixation post immunostaining

5) Padlock probe hybridization and ligation
6) RCA amplification
7) Imaging
Strip / Hybridize fluo probe / Wash / Image

**b** Initial NeuN Immunostaining

**c** Signal Intensity (a.u.)

**d** NeuN — GFAP

**e**

**f** Cassini · HCR

**g**

regions, where transcript density shifts from high to low (Fig. 1g, top). Along the hippocampal axis, both Cassini and HCR-FISH methods yielded comparable values, further validating that gene counts remain unaffected by expression density (Fig. 1g and Supplementary Fig. 10).

We next assessed Cassini's ability to simultaneously probe mRNA alongside both conventional and oligo-conjugated

immunostaining. Conventional immunostaining was performed on a mouse brain hemisphere to map the cellular localization of Nestin, a marker for neural progenitor cells, and gephyrin, a scaffolding protein associated with inhibitory synapses. Nestin is a cytoskeletal intermediate filament protein that forms expansive fibrous networks. By contrast, gephyrin displays a more uniform distribution, characterized by widespread punctae indicative of its localization to

**Fig. 1 | Cassini: evaluating RNA and protein detection performance. a** Overview of Cassini. **b** Effect of Cassini on conventional immunostaining and signal retention after to 2nd fixation. Microscope images display immunostaining prior to Cassini processing, with the white outlines indicating the regions selected for signal retention quantification shown in bar charts below. Dot plots represent biological triplicates for each condition. **c** Buffer optimization for conjugated antibodies. The microscope images show a zoom area of the CA1 hippocampal field. The white lines indicate the areas measured for signal intensity, which are plotted on the left. All conjugated immunostaining utilized the same lookup table (LUT) settings, while the conventional immunostaining images were enhanced to match the intensity levels of the conjugated staining. **d** Comparison of conventional and oligo conjugated immunostaining after Cassini run. The DAPI signal is shown in blue in both conjugated and regular antibody images, but not in the merged image to facilitate visualization. The lookup table of conventional and conjugated immunostaining have been adjusted to match the intensity. **e** Violin plot of foci area of HCR and Cassini. Bars in the violin plots indicate the standard deviation of the data. Violin plots were generated from selected areas with low foci density, aggregated from biological triplicates for each gene. **f** Comparison of foci density between Cassini and HCR. Density is the average number of foci around each dot in a 10 μm radius. At the top left is a schematic of the density calculation, and at the bottom left are the different genes and areas that were considered. Dot plots represent biological triplicates for each condition (except for *Cux2* Cassini series with only two replicates). Asterisks (*) indicate $p < 0.05$ without correction for multiple testing (one-tailed Student's *t* test with df = 4). Prox1, $p = 0.0152$; Slc17a7-CA-Lateral, $p = 0.048$; and Slc17a7-CA-Medial, $p = 0.038$. **g** Cassini/HCR ratios along the *Amigo2* gradient. The sliding window was 10% of the total length with an increment of 2%. The blue line indicates the average of all possible Cassini/HCR ratios (points) from triplicates experiments calculated for each increment. Note: Cassini data in **e**–**g** do not involve immunostaining and are limited to RNA detection.

inhibitory synapses. In this configuration, one channel is dedicated to DAPI (405 nm), two channels are used for conventional immunostaining (488 nm and 561 nm), and the remaining channel (638 nm) is employed for the detection of mRNAs and conjugated antibodies. On this same sample, we also measured expression of 8 RNA species, and stained with two conjugated antibodies in 10 rounds (Supplementary Fig. 11). The spatial patterns of the RNA features were highly concordant with the patterns of the same genes in Allen Mouse Brain Atlas[26] (Supplementary Fig. 10b). We confirmed that the conventional immunostaining (Nestin and Gephryn) remained intact throughout the 10 rounds of stripping and re-probing (Supplementary Fig. 12). This experiment demonstrates the ability of Cassini to retain conventional immunostaining patterns while also being able to measure gene and protein expression in a highly multiplexed, barcode-driven fashion.

Protein multiplexing was also performed on human tissue. Prior to analysis, we validated that the signal from the conjugated antibody showed strong correlation with conventional immunostaining (Supplementary Fig. 13). We multiplex five features (2 conventional + 3 conjugated) on pathological and healthy control tissues. As expected, β-amyloid plaques were observed in Alzheimer's tissue but were absent in the control. Additionally, we confirmed proper staining of white matter (NeuN and MAP2), vasculature (Ulex), and astrocytes (GFAP) in both tissue types (Supplementary Fig. 14).

Without conventional immunostaining, three channels can be used to detect different features in each round. In a final experiment to demonstrate Cassini's ease and throughput, we surveyed 30 mRNAs and 2 conjugated antibodies on a mouse brain hemisphere of ~40 mm² (Fig. 2 and Supplementary Fig. 15). The full buffer exchange and incubation time for this experiment was 47 min, with an imaging time across all four channels (405 nm−DAPI, 488 nm−Feature 1, 561 nm−Feature 2 and 638 nm−Feature 3) of 7 min, achieving a detection rate of 18 min per feature (Detailed workflow and protocol in supplementary material). The hippocampal regions were well delineated by *Prox1* (labeling the dentate gyrus), *Prdm8* (marking the CA2 and CA3 fields), *Amigo2* (a CA2-specific marker) and *Wfs1* (specifically expressed in CA1) (Fig. 2b). In addition, 11 neuronal marker genes for cortical layers were accurately stratified in the Cassini data (Fig. 2c, d). The neuronal markers showed strong enrichment around the hippocampus and the cortical layers, but were almost absent from white matter (*Mbp*). Moreover, markers of inhibitory neurons (*Gad2*, *Cnr1* and *Sst*) clustered together, as expected (Fig. 2d). The two conjugated antibodies we included in this run, NeuN and GFAP, demonstrated robust colocalization with their corresponding mRNA signals (Fig. 2e) as well as a strong correlation, especially with NeuN/*Rbfox3* (Supplementary Fig. 16). Together, these results demonstrate how highly multiplexed imaging of gene and protein expression, across a wide tissue area (40 mm²), can be performed in rapid, routine experiment.

## Discussion

We believe that for an open-source method to be truly effective, it must not only be easy to implement but also provide a clear protocol and easy design. All non-enzymatic reagents can be prepared in a few minutes from standard lab chemicals (formamide, SSC, Dextran sulfate), while enzymatic reactions utilize ready-to-use commercial kits. In addition to a clear protocol (Detailed workflow and protocol in supplementary material) we provide an online platform for designing probes (cassini.me) for 10 species.

In summary, we introduce Cassini, a method for the efficient, cost-effective surveying of proteomic and genomic features within intact tissue. Our approach performs multiplexing via sequential detection, enabling a simple setup and achieving a rate 50 times faster than state-of-the-art commercial techniques (multiplexed v3.HCR-FISH), without compromising sensitivity. Moreover, the total cost for detecting 30 genes and 2 conjugated antibodies, including all necessary probes, antibodies, reagents for Padlock amplification, and multiple rounds of fluorescent detection, was under $50 for this sample with manual buffer exchange and under $100 with automated buffer exchange (Supplementary Data 1). The ability of Cassini to rapidly survey dozens of transcript and protein types has broad applications in molecular biology, including cell type abundance profiling[27], monitoring infection across tissue, characterizing tumor microenvironment[28] and in situ analysis of perturbation assays[29]. At this level of throughput, ease, cost-efficiency and versatility, Cassini has the potential to become a routine tool for investigating tissue molecular organization.

## Methods
### Animal handling
All procedures involving animals at the Broad Institute were conducted in accordance with the US National Institutes of Health Guide for the Care and Use of Laboratory Animals under protocol number 0120-09-16. Wild-type C57BL/6 mice male (Charles River Laboratories) were housed in a 12-h light/12-h dark cycle with controlled temperature and humidity and ad libitum access to food and water. Mouse were scarified between p60 and p80.

### Human brain tissue
Tissue samples from Alzheimer and control cases were obtained from the Massachusetts Alzheimer's Disease Research Center (MADRC) at Massachusetts General Hospital. Brain tissues from selected donor autopsies performed at Massachusetts General Hospital are accompanied by the reports generated by the Neuropathology Core of the MADRC. These reports include neuropathology diagnosis of Alzheimer associated changes and descriptive clinical information on the severity of neurological signs and symptoms of Alzheimer associated cognitive decline. The study subjects and their next of kin gave written informed consent for the

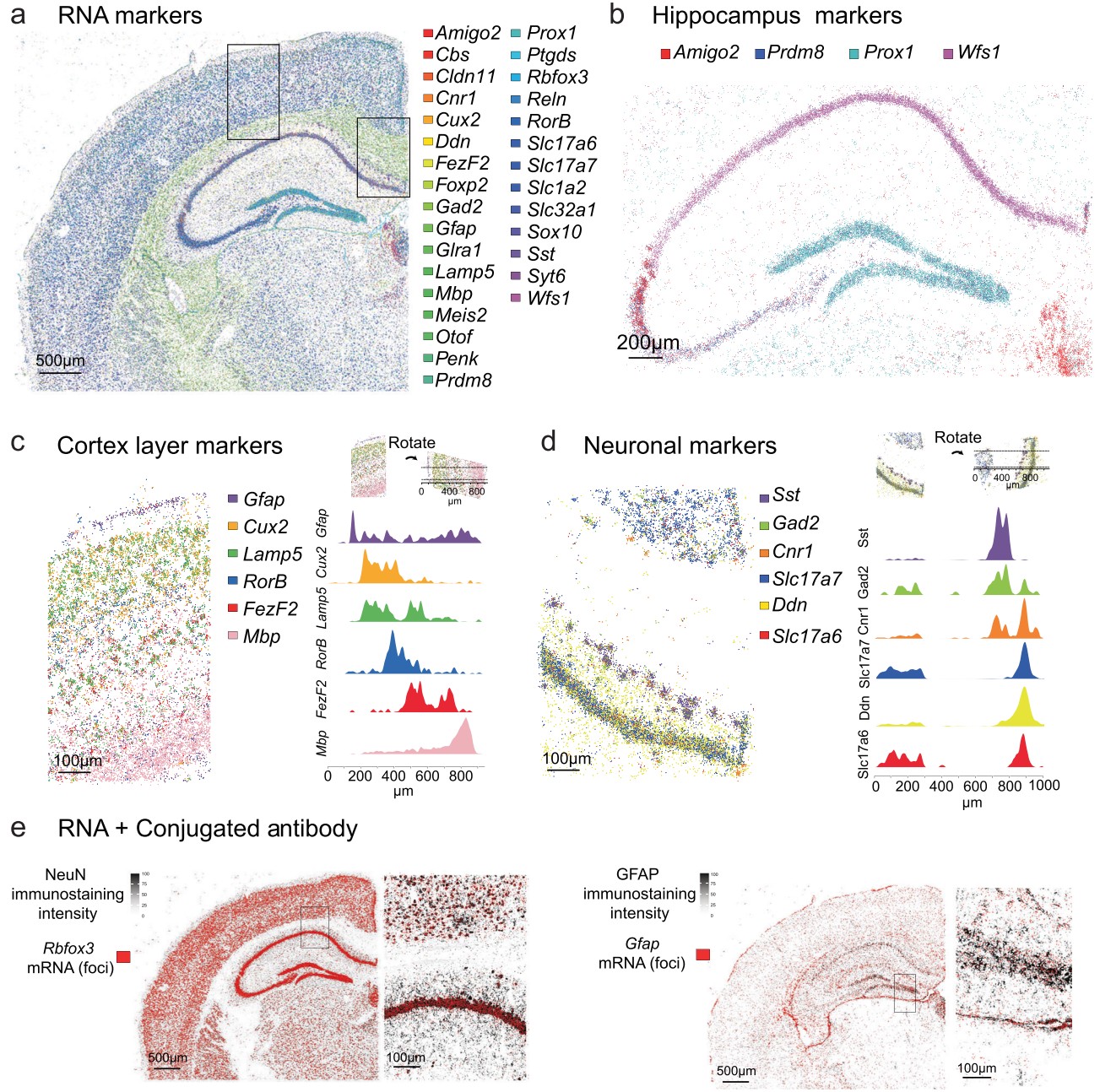

**Fig. 2 | Multiplexed detection of 30 RNAs and two conjugated antibodies. a** All 30 RNA types are displayed in a mouse brain hemisphere, with rectangles indicating zoomed-in areas shown in (**c**) and (**d**). **b** Four marker genes of the hippocampus. **c** Six markers indicating the cortical layers and **d** six neuronal markers, along with the corresponding density. In **c** and **d**, the density of the markers within the stripe indicated at the top has been calculated using the Epanechnikov kernel. **e** immunostaining (black) and the corresponding RNA (red) of GFAP and NeuN.

brain donation, and the Massachusetts General Hospital Institutional Review Board approved the study protocol. The subjects fulfilled the National Institute of Neurological and Communicative Disorders and Stroke, Alzheimer's Disease and Related Disorders Association criteria for probable Alzheimer and the National Institute on Aging-Reagan criteria for high likelihood of Alzheimer.

Control cortex comes from post-mortem autopsy from 94 years old male. Alzheimer cortex comes from post-mortem autopsy from 75 years old female. Use of the tissue at the Broad Institute was approved by the Office of Research Subject Protection project NHSR-4235. The samples were stored at −80 °C until use after equilibration at −20 °C in the cryostat.

**Transcardial perfusion**

C57BL/6 mice aged between postnatal day 60 (p60) and postnatal day 80 (p80) were anesthetized by administration of isoflurane in a gas chamber flowing 3% isoflurane for 1 min. Anesthesia was confirmed by checking for a negative tail pinch response. Animals were moved to a dissection tray, and anesthesia was prolonged via a nose cone flowing 3% isoflurane for the duration of the procedure. Transcardial perfusions were performed with ice-cold pH 7.4 HEPES buffer containing 110 mM NaCl, 10 mM HEPES, 25 mM glucose, 75 mM sucrose, 7.5 mM MgCl2 and 2.5 mM KCl to remove blood from the brain and other organs sampled. The brains were removed, frozen for 3 min in liquid nitrogen vapor and moved to −80 °C for long-term storage.

## Tissue handling

Before tissue slicing, 40 mm glass coverslips (Bioptechs Inc) were coated with Poly-L-Lysine Solution (0.01%) (Sigma A-005-C) for 30 min. The coverslips were then rinsed three times with Ultra pure H2O and dried. On each slides a 9x9mm hybridization chamber (Biorad SLF0201) was applied. Fresh frozen tissue was warmed to −18 °C in a cryostat (Leica CM3050S) for at least 15 min prior to handling. The tissue was then mounted onto a cutting block with OCT and sliced at a thickness of 10 μm. The tissue slices were deposited on the coated glass coverslips in the hybridization chamber and melted with a finger. The sliced tissue was kept inside the cryostat until processed for fixation and permeabilization.

## PCR assay

The tissue was stained with the conjugated NeuN antibody (see "Antibodies conjugation" section) in a conventional immunoblocking buffer (see "Immunostaining" section). Samples was treated with the Cassini protocol and stopped at the indicated points (after immunostaining, after probe hybridization, after ligation, after RCA). At the stopping point the tissue was digested using the digestion buffer described in ref. [30]. Digested tissue was cleaned with 2x spri and 1.6 x isopropanol. Samples were amplify with kapa hotstart readymix (Roche # KK2601) using ATAACTACATAATGATTA and CACGAGTGAACGAGAC primers and 62 °C annealing temperature for 12 cycles. Samples were diluted to 1:1000, 1:100, and 1:10 before being loaded onto a BioAnalyzer chip (Agilent). Quantification was perform from BioAnalyzer profile.

## Fixation and permeabilization

To each slide 50 μl of paraformaldehyde 4% in PBS (Thermo Fisher Scientific J61899.AK) and incubate for 12–14 min at room temperature and wash 3 times with 80 μl of PBS. Quench the PFA with 50 μl of 100 mM Tris-HCL pH 7.5, incubate 10 min at room temperature and wash 3 times with 80 μl of PBS. The sample was incubated with 50 μl of 0.25% Triton X (diluted in ultra pure water from stock solution Sigma X100-5ML) for 10 min at room temperature and wash 3 times with 80 μl of PBS. The sample was incubated 50 μl of 0.1 M HCl for 5 min at room temperature and wash 3 times with 80 μl of PBS.

## Antibodies conjugation

Anti-GFAP antibody (Synaptic System #173011) and anti-Neun antibody (Synaptic System #266008) and β-Amyloid (Cell signaling #9888T) were conjugated using oYo-Link® Oligo Custom system from AlphaThera. A total of 12.5 μl of antibodies (1 μg/μl) were mixed with 12.5 μl of oYo-Link reagent and crosslinked for 2 h using an LED PX2 Photo-Crosslinking Device (AlhaTherra) according to the manufacturer's instructions. The samples were then moved to 4c for storage. The sequences of the conjugated oligos are shown in Supplementary Data 3.

## Antibodies

In mouse tissue we used Anti-Nestin antibody (Abcam ab134017), Anti-Gephyrin antibody (Abcam ab181382 Lot GR3271026-4), anti-GFAP antibody (Synaptic System #173011) and anti-NeuN mouse antibody (Synaptic System #266008, Lot 1-4)

In human tissue we used anti-MAP2 antibody (Abcam ab92434), anti-NeuN human antibody (Proteintech #26975-1-AP), anti-β-Amyloid (Cell signaling #9888T), ULEX-Biotin (VectorLabs B-1065-2) and anti-GFAP antibody (Synaptic System #173011).

We used following secondary antibodies Donkey anti-Rabbit IgG AF647 (Thermo Fisher Scientific A-31573), Goat anti-Mouse Ig AF568 (Thermo Fisher Scientific A11031) and Goat anti-Chicken IgY AF488 (VWR #102649-304)

## Immunostaining

All steps were done in a humidified chamber. The blocking solution for oligo-conjugated antibody was prepared as followed: 1% of FcR Blocking Reagent mouse (Miltenyi biotec #130-092-575) or or 1% of FcR Blocking Reagent human (Miltenyi biotec #130-059-901), 50 mM Nacl, 1:100 RNAse inhibitor (Lucigen #30281-1-LU), 10% of ultra pure BSA 5% (Invitrogen #56773), 1% of Dextran sulfate-4K (Sigma #75027) completed with PBS.

The blocking solution for regular immunostaining was prepared as followed: 1% of FcR Blocking Reagent mouse (Miltenyi biotec #130-092-575) for mouse tissue 1:100 RNAse inhibitor (Lucigen #30281-1-LU), 10% of ultra pure BSA 5% (Invitrogen #56773) completed with PBS.

The blocking solution for oligo-conjugated antibody incompatible with RCA amplification was prepared as followed: 1% of FcR Blocking Reagent mouse (Miltenyi biotec #130-092-575) 1:100 RNAse inhibitor (Lucigen #30281-1-LU), 10% of ultra pure BSA 5% (Invitrogen #56773) 1% of high molecular dextran sulfate (Sigma #S4031) completed with PBS.

The samples were incubated 30 min at 4 °C with 50 μl of blocking solution. All non/conjugated antibodies were diluted in blocking buffer at a final concentration of 1:500. Conjugated antibodies were diluted in blocking buffer at a final concentration of 1:100,000. Fifty μl of the antibody solution was added to the samples and incubated overnight at 4 °C. Wash 3 times with 80 μl of blocking buffer, with 5 min incubation at room temp for each wash. Wash 3 times with 80 μl PBS and add 80 μl of PBS supplemented with 1:100 RNAse inhibitor (Y9240L Qiagen) and incubate 30 min at room temp. For conventional antibody staining, the secondary antibody was applied at a 1:500 dilution in 5% normal serum matched to the host species of the primary antibody. Except when mentioned otherwise, the samples were fixed a second time with 50 μL of 4% PFA (Thermo Fisher Scientific J61899.AK) and incubated for 2 h at room temperature. The PFA was removed and quenched with 50 μl of 20 mM Tris-HCL incubated 10 min at room temperature. Wash 3 times with 80 μl of PBS, with 5 min incubation at room temp for each wash.

## PLA based immunostaining

The PLA method used in Weibrecht et al.[18] was performed as followed: A blocking buffer was prepared with 2.5 ng/ul of Sonicated salmon sperm, 2.5 mM L-Cysteine (Sigma #168149-2.5G) diluted in Protein Block, Serum-Free (Agilent X090930-2). Sample was incubated 30 min at room temperature with 50 μl of blocking solution. Conjugated NeuN antibody (see "Antibodies conjugation" section) was diluted 10,000x in PBS and 10 times in blocking buffer. Blocking solution was removed and sample was incubated with 50 ul of antibody solution overnight at 4 °C. Sample were washed 1 time in 10 mM Tris-HCl with 0.1% Tween-20 for 5 min at room temperature and 2 times with PBS for 15 min and the sample was ready for probe hybridization.

The PLA method from Duolink In Situ PLA was run using Probe Anti-Rabbit commercial kit (Sigma DUO92002-30RXN) and performed as followed: Sample were incubated with 50ul of Duolink blocking solution for 1 h at 37 °C. Conjugated NeuN antibody (see "Antibodies conjugation" section) was diluted 10,000x in PBS and 10x in Duolink antibody diluent, for a final dilution of 100,000x. Blocking solution was removed and sample was incubated with 50 ul of antibody solution overnight at 4 °C. Sample were washed 1 time in 10 mM Tris-HCl with 0.1% Tween-20 for 5 min at room temperature and 2 times with PBS for 15 min and the sample was ready for probe hybridization.

## Ulex staining

ULEX-Biotin (VectorLabs B-1065-2) were mixed in the blocking buffer together at final concentration of 4 μg/ml. The tissue was wash as described in "Immunostaining" section and tissue was incubated 30 min with 4 μg/ml of Streptavidin (NEB N7021S) in PBS. The sample

was washed again 3 x times 15 min with PBS. Sample was incubate for 30 min with 100 nM of OL10 or OL11 (Supplementary Data 4) in PBS solution supplemented with 1% of Dextran sulfate-4K (Sigma #75027). Sample was washed 3 times 15 min with PBS solution supplemented with 1% of Dextran sulfate-4K (Sigma #75027). In Supplementary Fig. 13 we used the fluorescent oligo OL11 detected by probe OL13 to show dual staining "conventional" and conjugated. In Supplementary Fig. 14 we used oligo OL10 detected by probe OL13.

### Probe hybridization, ligation and RCA amplification

All steps were done in a humidified chamber and all temperature-controlled incubations were done in 37 °C in HybEZ™ II Oven (ACD-Bio). Wash-20 buffer contains 20% formamide (Sigma-Aldrich F7503-4L), 2X SSC (Thomas Scientific C000A15) in ultra pure water. The samples were preincubated with 50 µl of Wash-20 supplemented with 0.4 unit/µl of RNAse inhibitor (Lucigen #30281-1-LU) for 15 min at room temperature. The probe solution contains 10 nM of each probe 2X SSC, 20% formamide and 0.8 U/µl of RNAse inhibitor (Lucigen #30281-1-LU) in ultra pure water. The samples were incubated with 30 µl of probe solution overnight at 37 °C. After probe incubation, samples are washed 3 times with 80 µl of Wash-20 at 37 °C for 15 min each, followed by one wash with 80 µl of PBS for 15 min at 37 °C, followed by one wash with 80 µl of 1X splintR ligase buffer (NEB M0375L) for 15 min at room temperature. The ligase solution was prepared on ice, with 1x splintR ligase buffer (NEB M0375L), 1.25U/µl of ligase (NEB M0375L) and 0.8 U/µl of RNAse inhibitor (Lucigen #30281-1-LU). The samples were incubated with 50 µl of ligase solution and incubate 6 h to overnight at 37 °C The ligation was stopped with a wash of 80 µl of 2X SSC buffer incubated for 30 min at RT. The RCA primer hybridization mix was prepared by diluting in Wash-20 the RCA primer to a final concentration of 500 nM supplemented with 0.4 U/µl of RNAse inhibitor (Lucigen #30281-1-LU). The samples were incubated with 50 µl of RCA primer for 2 h at 37 °C). The sample was washed with 80 µl of Wash-20 incubated for 30 min at 37 °C, followed by a wash of 80 µl of PBS incubated for 15 min at 37 °C, followed by 80 µl 1X Phi-29 buffer (Thermo Fisher Scientific EP0094) incubated for 15 min at room temperature. The RCA mix was prepared on ice, with 1X Phi-29 buffer (Thermo Fisher Scientific EP0094) + 250 µM of dNTP (NEB N0447L) + 0.8U/µl of Phi29 (Thermo Fisher Scientific EP0094) + 0.8 U/µl of RNAse inhibitor (Lucigen #30281-2-LU). The samples were incubated with 50 µl of RCA mix and incubate overnight at 30 °C. After the RCA amplification, RCA mix was replaced with 80 µl of PBS. From this point the samples is stable for at least 3 months at 4 °C (probably more but we did not test it).

### Automated buffer exchange

The buffer exchange was performed using a FCS2 flow cell (Bioptechs), a pair of 12-inlets rotary valves (Advanced Microfluidic) and a peristaltic pump (Kamoer KCM-ODM-B253). The rotary valves and the peristaltic pumps were controlled by a Raspberry Pi using a script written in Python provided by the rotary valve manufacturer (AMF) and custom made for the pump control. The two rotary valves are connected in series. The first valve receives the probe mixes (one mix per cycle), and its outlet is connected to the inlet of the second rotary valve. Additionally, the inlets of the second rotary valve are connected to the stripping buffer, tissue blocking + DAPI buffer, and wash-20. The outlet of the second rotary valves is connected to the flow cell. The peristaltic pump is placed downstream of the flow cell and controls the flow rate at 0.3 ml/min for the probe mix buffer and 1 ml/min for the other buffers.

The probe mix contains 300 nM of each probe diluted in Wash-20; the stripping buffer contains 80% of formamide (Millipore F7503-4L) + 0.5X SSC (Thomas Scientific C000A15) diluted in distilled water; the tissue blocking + DAPI buffer contains 0.5% of High molecular weight (MW > 500,000) dextran sulfate (Sigma S4031) + 1:4000 DAPI

(Thermo Fisher # 62248) diluted in Wash-20. The cycle was run as follow. Stripping phase: flow 2.5 ml of stripping buffer incubate for 30 s, flow 1 ml of stripping buffer incubate for 30 s and flow 2.5 ml of stripping buffer incubate for 5 min. Tissue blocking and Dapi phase: flow 2.5 ml of tissue blocking + DAPI buffer incubate for 30 s and flow 2.5 ml of tissue blocking + DAPI buffer incubate for 3 min. The pre-incubation phase: flow 2.5 ml of Wash-20 incubate for 30 s and flow 2.5 ml of Wash-20 and incubate for 2 min. Probe incubation phase: Flow 0.8 ml of probe mix and incubate for 25 min. Probe wash phase: Flow 2 ml of Wash-20 incubate 1 min, repeat 5 times flow 1 ml of Wash-20 incubate 1 min and flow 3 ml of Wash-20 incubate 30 s. Imaging of the sample: The imaging automation was achieved using a mouse autoclicker that clicked on the microscope interface at regular intervals.

### Imaging system

Imaging was performed using Nikon Eclipse Ti2 inverted confocal microscope equipped with an Andor Dragonfly (Oxford Instruments) with a 40 um single pinhole spinning disk unit and a Zyla Plus 4.2 Megapixel CMOS camera. We used 405 nm, 488 nm, 561 nm and 638 nm laser lines paired with 445/46, 521/38, 594/43, 684/47 emission filters, respectively. All images were done using a 20x objective (NA 0.75, Nikon). Microscope control, acquisition, and stitching were performed using the Fusion interface.

### Probe design and ordering

The probes were designed from the exon sequences of the genes using the GRCm39 mouse genome assembly and are available in Supplementary Data 2. The gene sequences were scanned with a sliding window of 32 nucleotides to find a sequence that match the following criteria: the Tm of each arm (16 nt) is >50 °C and the difference of Tm between 5' and 3' arms is <9 °C. The GC range of the 32 nt binding sequence (5' + 3' arms) is between 45% and 65% and the probe junction with GC, GG or CG were excluded. The 5' to 3' sequence of the probe is composed of the 5' arm, RCA primer binding sequence, detection probe binding sequence, spacer and 3' arm. We provide an online platform, cassini.me, for designing exonic or intronic probes for almost all human and mouse genes.

The RNA probes (Supplementary Data 2) were ordered as an oPool from IDT with a 5'phosphate group. The probe targeting the oligos on the conjugated antibodies (Supplementary Data 4) were ordered as individual oligos from IDT with a 5'phosphate group. The RCA primers (Supplementary Data 4) were ordered from IDT, with two phosphorothioate bounds on the 3' to prevent exonuclease activity from Phi29. The fluorescent probes (Supplementary Data 3) were ordered from Gene Link. The fluorophore was placed on the 5' of the probe sequence and in most fluorescent probes two adjacent thymines were added between the fluorophore and the binding sequence to prevent any nucleobase-specific quenching. Fluorescent probes were purified by the manufacturer using reverse cartridge.

### Image and data processing

Alignment: the set of microscopy images from multiple rounds of hybridization and stripping were aligned on the Dapi channel of round 0 using a Python script. In brief the intensity was rescaled and translation was applied for image registration. The transformation matrix was computed using *register* function from the *pystackreg* package. For each channel the transformation matrix is applied using "transform" function from the *pystackreg* package and the aligned image is saved in tiff format.

**Foci identification.** The foci were identified using the "Find Maxima" function in FIJI (https://imagej.net/software/fiji/), with the prominence parameters set to 200 for the Cassini images and 40 for the HCR images (Supplementary Fig. 4).

**Local density.** The foci information from "Find Maxima "function in FIJI were imported and an area of interest was selected. For each foci in the area of interested, a circle with a radius of 10 μm was drawn, and the number of foci within this circle was counted.

**Segmentation.** The "Find Maxima" function in FIJI was applied to the Dapi channel of the microscopy images. Segmentation data were then generated using the "Segmented Particles" function and saved in.roi format. The.roi files were converted to coordinates using python package *read_roi*. Foci and protein signals were segmented using an R script that iteratively processed each region of interest (ROI). Protein and RNA signals within each ROI were quantified using the *point.in.-polygon* function.

## HCR-FISH
Three different sets of probes were used for HCR-FISH (Molecular Instruments). Set 1 composed of standard probe of *Gfap* B1-488, *Prox1* B2-647 and Slc17a7 B3-546; Set 2 composed standard probe *Mbp* B3-488 and *Rbfox3* B4-647. Set 3 composed of custom probe *Wfs1* B1-488, *Cux2* B2-647 and *Amigo2* B3-546. The protocol was followed according to the manufacturer's instructions.

## Statistics and reproducibility
Statistics shown in Fig. 1f is specified in the figure legend. Data shown Fig. 1b, e, f, g are from biological triplicates. Microscopy pictures shown in Fig. 1c are representative micrographs from triplicate experiments. Microscopy pictures shown in Fig. 1d and microscopy data in Fig. 2 are representative data from duplicate experiments. All raw data used in charts with replicates are available in source_data.xlsx file.

## Reporting summary
Further information on research design is available in the Nature Portfolio Reporting Summary linked to this article.

## Data availability
All data supporting the findings of this study are available within the paper and its Supplementary Information. The raw data associated with the figure graphs can be found in the Source Data file. The raw microscopy image data generated in this study has been deposited in the Zenodo database. Raw data of Fig. 1 and Supplementary Figs. 2, 3, 5, 6 can be found under accession codes https://doi.org/10.5281/zenodo.16892950 (https://zenodo.org/records/16892950). Raw data of Supplementary Figs. 1, 13 and 14 can be found under accession codes https://doi.org/10.5281/zenodo.16893769 (https://zenodo.org/records/16893769). Raw data of Fig. 2 and Supplementary Figs. 8, 10, 15 and 16 can be found under accession codes https://doi.org/10.5281/zenodo.16885814 (https://zenodo.org/records/16885814); Raw data of Supplementary Figs. 4, 11 and 12 can be found under accession codes https://doi.org/10.5281/zenodo.16878401 (https://zenodo.org/records/16878401). Raw data of Supplementary Figs. 8 and 10 for Cassini can be found under accession codes https://doi.org/10.5281/zenodo.16891291 (https://zenodo.org/records/16891291). Raw data of Supplementary Figs. 8 and 10 for HCR can be found under accession codes https://doi.org/10.5281/zenodo.16891465 (https://zenodo.org/records/16891465) Source data are provided with this paper.

## Code availability
The code is available at https://github.com/Cassini-insitu/image_processing (also check https://doi.org/10.5281/zenodo.16786976).

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

## Acknowledgements

We thank Fei Chen and the member of the Fei Chen lab at the Broad Institute for sharing their expertise on RCA-based mRNA detection and for providing access to their microscopy facility. We thank all members of the Macosko lab for feedback and discussions, especially Charles R. Vanderburg for providing and preparing the human brain tissue. This research was supported by the Swiss National Science Foundation Early Postdoc.Mobility fellowship (P2EZP3_184261) and the National Institutes of Health (NIH) grant 1U19MH114821 (Macosko).

## Author contributions

N.L. and M.T.K. conceptualized and designed the experiments under the supervision of E.Z.M. N.L. and M.T.K. performed most of the experiments, with tissue slicing primarily carried out by N.N., and the different fixation conditions in Fig. 1b prepared by N.T. and J.P. N.L. developed the data analysis and visualization code. N.L. drafted the manuscript with contributions from all authors.

## Competing interests

The authors declare no competing interests.
