## [Transparent Peer Review file · Nature Communications]

Cassini: A scalable, multiplexed approach to joint RNA and protein profiling in situ

Corresponding Author: Dr Evan Macosko

Version 0:

Reviewer comments:

Reviewer #1

(Remarks to the Author)

The authors present a method called Cassini, which involves padlock probe- and immune-RCA mediated combined RNA and protein detection. The study is very well conducted and data seem solid and accurately analyzed. The manuscript is generally well written but the citations are missing important prior works.

The main problem is that the method presented in the manuscript has very little novelty. The multiplexed mRNA detection approach is identical to SCRINSHOT (Sountoulidis, A., et al. PLoS Biol 18(2020)), both employing SplintR driven multiplex padlock probe ligation on mRNA followed by a non-combinatorial serial detection scheme, identical to the one presented. The protein detection method is identical to Immuno-RCA (Schweitzer et al 2000 <https://www.pnas.org/doi/10.1073/pnas.170237197>). Immune-RCA kits and reagents (including blocking buffer) available from Sigma as Duo-Link products. The combination of padlock-RCA and immune-RCA is not novel either (Weibrecht I. et al PLoS One. (2011) 6, e20148 & Weibrecht I, et al. Nat Protoc. (2013) 8, 355-72. Oligo-conjugated antibodies have been successfully been used in combination with RCA with the blocking reagents provided in the publications (thousands of publications), thus the added value of the proposed blocking buffer is unclear.

Given that none of these prior works have been cited, one might get the impression that the method is more novel than it is. The manuscript needs to be revised to make a better account of prior works, and tune down the claims of novelty.

Reviewer #2

(Remarks to the Author)

Spatial omics technology presents significant challenges, even in reproducing existing methods. Optimizing each step to enhance the reproducibility and efficiency of experimental systems remains a critical issue. In particular, the detection of target proteins using oligo DNA-conjugated antibodies in combination with RCA often involves a trade-off between detection specificity and sensitivity. The authors aimed to address this challenge by optimizing conditions during immunostaining, specifically refining the blocking buffer, to achieve both the specific binding of oligo DNA-conjugated antibodies to targets and efficient RCA enzymatic reactions. Furthermore, they demonstrated the simultaneous detection of proteins and RNA while preserving spatial information by integrating RCA-based mRNA detection with this method.

While this study presents an interesting topic, many researchers have already reported simultaneous detection of proteins and RNA while maintaining spatial information. Consequently, the novelty of this work is insufficient. Below, I provide comments on the manuscript.

1. In Fig. 1a, the authors present an overview of Cassini. It would be beneficial to explicitly clarify which aspects of Cassini represent advancements over existing technologies to help readers better understand the unique features of this method.
2. While numerous methods for highly multiplexed simultaneous detection of proteins and RNA have been reported, the manuscript should clearly delineate what can be uniquely achieved with Cassini compared to these existing techniques.
3. The authors demonstrate the detection of up to 30 mRNAs and 2 proteins using Cassini. To better highlight the multiplexed measurement capability of this method, it would be more compelling to demonstrate its ability to detect a greater number of protein features (e.g., more than 4) compared to conventional immunostaining.

4. Demonstrating the capability for multi-round experiments with multi-color imaging would strengthen the manuscript. While the main text states that three channels are available per round, the multi-round images presented in Supplementary Fig. 9 only include single-color data for 10 rounds. Including multi-color images across multiple rounds would provide stronger validation of this capability.

5. There appears to be a potential inconsistency in the reported detection time. The abstract states that each feature can be detected in under 20 minutes, while the main text reports a detection rate of approximately 18 minutes per feature. However, Supplementary Protocol 37 indicates that each cycle requires 47 minutes plus 7 minutes for imaging, suggesting that the stated time may refer to per-color imaging in a three-color system. If this is the case, the total detection time per feature may be longer than implied. Clarifying this in the text would improve consistency and prevent potential misinterpretation.

Minor Comments

1. Gene names in Fig. 2 and the Supplemental Figures should be italicized.
2. In the imaging system description, "405n" should be corrected to "405 nm."
3. More details about the lens, particularly the numerical aperture (NA), should be included.
4. The readability of the Cassini Workflow in the Supplemental Figures should be improved.

Reviewer #3

(Remarks to the Author)

Cassini: Streamlined and Scalable Method for in situ profiling of RNA and Protein

In this study, the authors develop and demonstrate a scalable method for jointly measuring RNA and proteins in intact tissue sections at a fraction of the time and cost of currently available commercial methods. By combining rolling circle amplification and an optimized immunostaining buffer, Cassini achieves rapid detection (approximately 18 minutes per feature) and maintains sensitivity comparable to established methods like HCR.

The quantitative comparison of antibody signal between conjugated vs. conventional immunostaining (Fig. 1c) is compelling.

Overall, Cassini is a useful extension to the existing suite of spatial multi-omic methods based on its streamlined protocol and significant reduction in time and cost. The manuscript is well-written and concise.

Major points:

A major limitation of this work is the lack of validation on non-mouse brain tissues. As this team of authors likely knows well, extending in situ methods originally developed in the context of the mouse brain does not work readily in diverse tissues. Applying Cassini to other tissues with different autofluorescence properties, which could affect sensitivity and specificity, would be more compelling to position Cassini as a generalizable method.

Fig. 1d shows a speckled background noise in the conjugated NeuN antibody signal outside the dentate gyrus, suggesting that further refinement of the staining conditions may enhance signal specificity. A quantification of how such off-target binding confounds accurate localization of protein expression would be helpful, especially in tissues outside of the mouse brain.

Version 1:

Reviewer comments:

Reviewer #1

(Remarks to the Author)

My comments have been appropriately addressed in the revised version of the manuscript, and I now think it is acceptable for publication.

Reviewer #2

(Remarks to the Author)

Additional Comments

Thank you for the thorough and mostly convincing revisions. I am satisfied with the authors' responses and the new data provided for Comments 1–4 and 6, as well as the minor points. The manuscript is now much clearer, and the added experiments substantially strengthen the paper.

Remaining issue (Comment 5 – multi-round, multi-colour imaging):

Demonstrating true multi-round and multi-color capability with raw or minimally processed images is essential for a rigorous technical evaluation and is considered standard in the spatial omics community. Currently, Supplementary Fig. 11 and Fig. 2

describe how the channels are allocated; however, they do not display representative composite images from successive rounds that show the simultaneous detection of three distinct features within each round.

Reviewer #3

(Remarks to the Author)

The authors have satisfactorily addressed all comments.

We appreciate the time and effort the reviewers have dedicated to evaluating our manuscript **Cassini: Streamlined and Scalable Method for in situ profiling of RNA and Protein**. We are grateful for the thoughtful comments and suggestions, which have helped us improve the clarity and quality of our work.

Below, we address each comment point-by-point our answer is written in red. We have revised the manuscript accordingly.

REVIEWER COMMENTS

Reviewer #1 (Remarks to the Author):

The authors present a method called Cassini, which involves padlock probe- and immune-RCA mediated combined RNA and protein detection. The study is very well conducted and data seem solid and accurately analyzed.

“The manuscript is generally well written but the citations are missing important prior works.”

The main problem is that the method presented in the manuscript has very little novelty.

Comment 1

The multiplexed mRNA detection approach is identical to SCRINSHOT (Sountoulidis, A., et al. PLoS Biol 18(2020)), both employing SplintR driven multiplex padlock probe ligation on mRNA followed by a non-combinatorial serial detection scheme, identical to the one presented.

Answer 1

We were not previously aware of this reference. It is indeed highly relevant to our work and has now been included. However, we believe this does not compromise the novelty of our work, as our method is uniquely compatible with protein multiplexing. All existing blocking buffers we tested, including the PLA buffer you suggested, failed to yield specific single-protein immunostaining. This innovation, to our knowledge, is the first to maintain both the specific binding of oligo-conjugated antibodies and enzymatic activity, enabling seamless multimodal analysis.

In the revised version of the manuscript SCRINSHOT is cited as follows:

“Previous work demonstrated that RCA using SplintR ligase enables multiplexed mRNA detection(Sountoulidis et al. 2020), however this method is limited to RNA detection. To enable truly multimodal and multiplexed analysis using padlock probes, we devised: (a) a novel staining buffer to mitigate enzymatic inhibition while preserving specificity; and (b) a post-staining fixation protocol to ensure the retention of the signal.”

Comment 2

The protein detection method is identical to Immuno-RCA (Schweitzer et al 2000 <https://www.pnas.org/doi/10.1073/pnas.170237197>).

Answer 2

This method was developed for in vitro microarray detection, and its blocking buffer is not suitable for tissue immunostaining. Nonetheless, we acknowledge the importance of this pioneering work and have cited it accordingly:

' Existing methods have employed RCA for detecting antibody–DNA conjugates either in vitro (Schweitzer et al 2000), ...'

Comment 3

Immune-RCA kits and reagents (including blocking buffer) available from Sigma as Duo-Link products. The combination of padlock-RCA and immune-RCA is not novel either (Weibrecht I. et al PLoS One. (2011) 6, e20148 & Weibrecht I, et al. Nat Protoc. (2013) 8, 355-72.)

Answer 3

We acknowledge that those references to PLA are highly relevant and apologize for their initial omission.

We are now citing those references in two separate parts of the manuscript:

“Although some methods have demonstrated simultaneous detection of RNA and protein proximity using enzymatic amplification (Weibrecht et al. 2011 and Weibrecht et al. 2013), these are not applicable to conventional single-protein immunostaining.”

“Existing methods have employed RCA for detecting antibody–DNA conjugates either in vitro (Schweitzer et al 2000), or in proximity ligation assays (Weibrecht et al. 2011 and Weibrecht et al. 2013) typically using similar buffers containing salmon sperm DNA and protein blockers. However, both the commercial buffer (Duolink) and those reported in the literature (Weibrecht et al. 2013) exhibit substantially lower specificity compared to the Cassini buffer (Supp. Fig. 4).”

We wish to emphasize that the primary focus of these cited references is on using RCA amplification in the context of proximity ligation assays rather than for single-protein immunostaining. Supplementary Figure S4 (reproduced below) demonstrates that the PLA blocking buffer is not an effective choice for single-protein NeuN immunostaining:

Comment 4

Given that none of these prior works have been cited, one might get the impression that the method is more novel than it is. The manuscript needs to be revised to make a better account of prior works, and tune down the claims of novelty.

Answer 4

All the prior works mentioned in the reviewer comments have now been incorporated into our revised manuscript. Nonetheless, we do not think their inclusion compromises the novelty of our method, as these previous approaches target either RNA alone or RNA combined with proximity ligation, but none of them integrates RNA detection with single-protein immunostaining.

Reviewer #2 (Remarks to the Author):

Spatial omics technology presents significant challenges, even in reproducing existing methods. Optimizing each step to enhance the reproducibility and efficiency of experimental systems remains a critical issue. In particular, the detection of target proteins using oligo DNA-conjugated antibodies in combination with RCA often involves a trade-off between detection specificity and sensitivity. The authors aimed to address this challenge by optimizing conditions during immunostaining, specifically refining the blocking buffer, to achieve both the specific binding of oligo DNA-conjugated antibodies to targets and efficient RCA enzymatic reactions. Furthermore, they demonstrated the simultaneous detection of proteins and RNA while preserving spatial information by integrating RCA-based mRNA detection with this method.

Comment 1

While this study presents an interesting topic, many researchers have already reported simultaneous detection of proteins and RNA while maintaining spatial information. Consequently, the novelty of this work is insufficient.

Answer 1

Our method is unique as it is the only microscopy-based approach that can survey RNA and proteins with high multiplexing capacity, single-protein immunostaining and strong amplification and fast readout. All other methods cited in our manuscript lack at least one of these key features.

Liu, Y. *et al.* High-plex protein and whole transcriptome co-mapping at cellular resolution with spatial CITE-seq. *Nat. Biotechnol.* **41**, 1405–1409 (2023). Is an NGS based method with very high multiplexing but low resolution and sensitivity.

He, S. *et al.* High-plex imaging of RNA and proteins at subcellular resolution in fixed tissue by spatial molecular imaging. *Nat. Biotechnol.* **40**, 1794–1806 (2022) → Is a complex combinatorial based detection method based on conventional ISH probe (limited amplification and specificity).

In situ detection of individual mRNA molecules and protein complexes or post-translational modifications using padlock probes combined with the in situ proximity ligation assay. *Nat. Protoc.* **8**, 355–372 (2013). → The system has some similarities to Cassini, but is designed to assay proximity ligation.

Hybridization chain reaction enables a unified approach to multiplexed, quantitative, high-resolution immunohistochemistry and in situ hybridization. *Dev. Camb. Engl.* **148**, dev199847 (2021)→ This approach lacks Cassini's multiplexing capabilities.

If you believe we are missing published papers that compromise the novelty of our study, we would be happy to discuss them and explore how we might address this.

Comment 2

1. In Fig. 1a, the authors present an overview of Cassini. It would be beneficial to explicitly clarify which aspects of Cassini represent advancements over existing technologies to help readers better understand the unique features of this method.

Answer 2

We appreciate this suggestion and have revised Figure 1a to better highlight all the key features which altogether are specific to our method.

- a) (3) demonstrates that LMW is an essential component of the blocking buffer, previously unknown to provide specificity without interfering with enzymatic reactions.
- b) In (6), we show RCA amplification on conjugated antibody. To the best of our knowledge, this is the first method to show that RCA amplification can be used in fixed tissue to enable specific protein immunostaining with conjugated antibodies.

- c) (7) has been updated to show that the cycling process can be repeated as many times as needed and completed very rapidly, highlighting the multiplexing capacity of Cassini.

Comment 3

2. While numerous methods for highly multiplexed simultaneous detection of proteins and RNA have been reported, the manuscript should clearly delineate what can be uniquely achieved with Cassini compared to these existing techniques.

Answer 3

In the second paragraph of the main text, which discusses microscopy-based technologies capable of analyzing both proteins and RNA, we have added additional references and clarified the limitations of existing approaches. These technologies typically fall into three categories: (1) those that are slow with low multiplexing capacity; (2) methods that rely on protein proximity assays rather than true single-protein detection; and (3) approaches based on conventional ISH probe with limited signal amplification and/or signal specificity.

In contrast, our technology, as outlined in the abstract, is *“a new approach for straightforward, cost-effective multiplexed measurements of mRNA and protein features simultaneously. Cassini leverages rolling circle amplification (RCA), known for its robust amplification and remarkable stability even after intense stripping, to serially detect each feature in under 20 minutes”*.

Comment 4

3. The authors demonstrate the detection of up to 30 mRNAs and 2 proteins using Cassini. To better highlight the multiplexed measurement capability of this method, it would be more compelling to demonstrate its ability to detect a greater number of protein features (e.g., more than 4) compared to conventional immunostaining.

Answer 4

We appreciate this comment and agree that demonstrating protein multiplexing is important to highlight Cassini's advantages. To address this, we conducted a new experiment showcasing the simultaneous detection of five protein markers in both control and pathological human brain tissue (Supplementary Fig. 13 and 14).

Validation of the immunostaining as be done using co-immunostaining of conventional and conjugated antibodies shown in Supplementary Fig. 13:

The protein multiplexed experiment is shown in Supplementary Fig. 14 and analyzed as follows:

“As expected, β -amyloid plaques were observed in Alzheimer's tissue but were absent in the control. Additionally, we confirmed proper staining of white matter (NeuN and MAP2), vasculature (Ulex), and astrocytes (GFAP) in both tissue types (Supp. Fig. 14).”

Comment 5

4. Demonstrating the capability for multi-round experiments with multi-color imaging would strengthen the manuscript. While the main text states that three channels are available per round, the multi-round images presented in Supplementary Fig. 9 only include single-color data for 10 rounds. Including multi-color images across multiple rounds would provide stronger validation of this capability.

Answer 5

We apologize for the confusion. The original Supplementary Fig. 9—which is now Supplementary Fig. 11 in our revised manuscript—shows two channels used for conventional immunostaining, while one channel is dedicated to sequential cycling

through RNA targets and conjugated antibodies. The data shown in the original Supplementary Fig. 9 (now Supplementary Fig. 11) are now stated as :

“In this configuration, one channel is dedicated to DAPI (405 nm), two channels are used for conventional immunostaining (488 nm and 561 nm), and the remaining channel (638 nm) is employed for the detection of mRNAs and conjugated antibodies.”

Figure 2 presents a distinct experiment that does not include conventional immunostaining, allowing all three channels to be used for detecting different features in each round. This is now clearly stated as:

“Without conventional immunostaining, three channels can be used to detect different features in each round. In a final experiment to demonstrate Cassini’s ease and throughput, we surveyed 30 mRNAs and 2 conjugated antibodies on a mouse brain hemisphere of ~40mm² (Fig. 2).”

Comment 6

5. There appears to be a potential inconsistency in the reported detection time. The abstract states that each feature can be detected in under 20 minutes, while the main text reports a detection rate of approximately 18 minutes per feature. However, Supplementary Protocol 37 indicates that each cycle requires 47 minutes plus 7 minutes for imaging, suggesting that the stated time may refer to per-color imaging in a three-color system. If this is the case, the total detection time per feature may be longer than implied. Clarifying this in the text would improve consistency and prevent potential misinterpretation.

Answer 6

We apologize for not stating this more clearly in our original submission. The full cycle takes 47 minutes, with an additional imaging time of all 4 colors (including DAPI) ranging from 1 to 7 minutes, resulting in a total cycle duration of approximately 48 to 54 minutes. Since we measure three features per cycle, this corresponds to a detection rate of roughly 18 minutes per feature (or <20 minutes) with long imaging time.

To make clear that the imaging time includes all 4 colors, we update the text as follows:

“The full cycle time for this experiment was 47 minutes, with an imaging time across all four channels (405 nm - DAPI, 488 nm - Feature 1, 561 nm - Feature 2 and 638 nm - Feature 3) of 7 minutes, achieving a detection rate of 18 minutes per feature (Detailed workflow and protocol in supplementary material)”

Minor Comments

1. Gene names in Fig. 2 and the Supplemental Figures should be italicized. **Done**
2. In the imaging system description, “405n” should be corrected to “405 nm.” **Done**
3. More details about the lens, particularly the numerical aperture (NA), should be included. **Done**
4. The readability of the Cassini Workflow in the Supplemental Figures should be improved. **Workflow labels have been repositioned for improved visibility, and a legend has been added for clarification**

Reviewer #3 (Remarks to the Author):

Cassini: Streamlined and Scalable Method for in situ profiling of RNA and Protein

In this study, the authors develop and demonstrate a scalable method for jointly measuring RNA and proteins in intact tissue sections at a fraction of the time and cost of currently available commercial methods. By combining rolling circle amplification and an optimized immunostaining buffer, Cassini achieves rapid detection (approximately 18 minutes per feature) and maintains sensitivity comparable to established methods like HCR.

The quantitative comparison of antibody signal between conjugated vs. conventional immunostaining (Fig. 1c) is compelling.

Overall, Cassini is a useful extension to the existing suite of spatial multi-omic methods based on its streamlined protocol and significant reduction in time and cost. The manuscript is well-written and concise.

Major points:

Comment 1

A major limitation of this work is the lack of validation on non-mouse brain tissues. As this team of authors likely knows well, extending in situ methods originally developed in the context of the mouse brain does not work readily in diverse tissues. Applying Cassini to other tissues with different autofluorescence properties, which could affect sensitivity and specificity, would be more compelling to position Cassini as a generalizable method.

Answer 1

We appreciate this comment, and agree that extending Cassini beyond mouse brain is important. To address this limitation, we applied Cassini to both pathological and control human cortex samples, which are well known for their high levels of autofluorescence (Supp. Fig. 13 and 14). We assayed a total of 5 antigens, across both control and

Alzheimer's affected brain. The results highlight our ability to distinguish these five protein epitopes with different distributions within the tissue and analyzed as follows:

“As expected, β -amyloid plaques were observed in Alzheimer's tissue but were absent in the control. Additionally, we confirmed proper staining of white matter (NeuN and MAP2), vasculature (Ulex), and astrocytes (GFAP) in both tissue types (Supp. Fig. 14).”

Comment 2

Fig. 1d shows a speckled background noise in the conjugated NeuN antibody signal outside the dentate gyrus, suggesting that further refinement of the staining conditions may enhance signal specificity. A quantification of how such off-target binding confounds accurate localization of protein expression would be helpful, especially in tissues outside of the mouse brain.

Answer 2

We agree that the background signal observed in the fiber tract is unexpected for NeuN immunostaining. To address this, we have added a new supplementary figure demonstrating that the signal is not due to antibody conjugation. We now explain this in the revised manuscript as follows:

“The speckled background observed in the fiber tract appears in both conventional and conjugated NeuN immunostaining (Fig. 1d left and Supp. Fig. 6), suggesting it is not specific to the conjugated method but likely reflects non-specific binding inherent to the antibody.”

Moreover, the quantification shown in Fig. 1c demonstrates the absence of background signal in the fiber tract located dorsal to CA1.

We appreciate the time and effort the reviewers have dedicated to evaluating our manuscript **Cassini: Streamlined and Scalable Method for in situ profiling of RNA and Protein**. We are grateful for the thoughtful comments and suggestions, which have helped us improve the clarity and quality of our work.

Below, we address each comment point-by-point our answer is written in red. We have revised the manuscript accordingly.

REVIEWER COMMENTS ROUND 1

Reviewer #1 (Remarks to the Author):

The authors present a method called Cassini, which involves padlock probe- and immune-RCA mediated combined RNA and protein detection. The study is very well conducted and data seem solid and accurately analyzed.

“The manuscript is generally well written but the citations are missing important prior works.”

The main problem is that the method presented in the manuscript has very little novelty.

Comment 1

The multiplexed mRNA detection approach is identical to SCRINSHOT (Sountoulidis, A., et al. PLoS Biol 18(2020)), both employing SplintR driven multiplex padlock probe ligation on mRNA followed by a non-combinatorial serial detection scheme, identical to the one presented.

Answer 1

We were not previously aware of this reference. It is indeed highly relevant to our work and has now been included. However, we believe this does not compromise the novelty of our work, as our method is uniquely compatible with protein multiplexing. All existing blocking buffers we tested, including the PLA buffer you suggested, failed to yield specific single-protein immunostaining. This innovation, to our knowledge, is the first to maintain both the specific binding of oligo-conjugated antibodies and enzymatic activity, enabling seamless multimodal analysis.

In the revised version of the manuscript SCRINSHOT is cited as follows:

“Previous work demonstrated that RCA using SplintR ligase enables multiplexed mRNA detection(Sountoulidis et al. 2020), however this method is limited to RNA detection. To enable truly multimodal and multiplexed analysis using padlock probes, we devised: (a) a novel staining buffer to mitigate enzymatic inhibition while preserving specificity; and (b) a post-staining fixation protocol to ensure the retention of the signal.”

Comment 2

The protein detection method is identical to Immuno-RCA (Schweitzer et al 2000 <https://www.pnas.org/doi/10.1073/pnas.170237197>).

Answer 2

This method was developed for in vitro microarray detection, and its blocking buffer is not suitable for tissue immunostaining. Nonetheless, we acknowledge the importance of this pioneering work and have cited it accordingly:

' Existing methods have employed RCA for detecting antibody–DNA conjugates either in vitro (Schweitzer et al 2000), ...'

Comment 3

Immune-RCA kits and reagents (including blocking buffer) available from Sigma as Duo-Link products. The combination of padlock-RCA and immune-RCA is not novel either (Weibrecht I. et al PLoS One. (2011) 6, e20148 & Weibrecht I, et al. Nat Protoc. (2013) 8, 355-72.)

Answer 3

We acknowledge that those references to PLA are highly relevant and apologize for their initial omission.

We are now citing those references in two separate parts of the manuscript:

“Although some methods have demonstrated simultaneous detection of RNA and protein proximity using enzymatic amplification (Weibrecht et al. 2011 and Weibrecht et al. 2013), these are not applicable to conventional single-protein immunostaining.”

“Existing methods have employed RCA for detecting antibody–DNA conjugates either in vitro (Schweitzer et al 2000), or in proximity ligation assays (Weibrecht et al. 2011 and Weibrecht et al. 2013) typically using similar buffers containing salmon sperm DNA and protein blockers. However, both the commercial buffer (Duolink) and those reported in the literature (Weibrecht et al. 2013) exhibit substantially lower specificity compared to the Cassini buffer (Supp. Fig. 4).”

We wish to emphasize that the primary focus of these cited references is on using RCA amplification in the context of proximity ligation assays rather than for single-protein immunostaining. Supplementary Figure S4 (reproduced below) demonstrates that the PLA blocking buffer is not an effective choice for single-protein NeuN immunostaining:

Comment 4

Given that none of these prior works have been cited, one might get the impression that the method is more novel than it is. The manuscript needs to be revised to make a better account of prior works, and tune down the claims of novelty.

Answer 4

All the prior works mentioned in the reviewer comments have been incorporated into our revised manuscript. Nonetheless, we do not think their inclusion compromises the novelty of our method, as these previous approaches target either RNA alone or RNA combined with proximity ligation, but none of them integrates RNA detection with single-protein immunostaining.

Reviewer #2 (Remarks to the Author):

Spatial omics technology presents significant challenges, even in reproducing existing methods. Optimizing each step to enhance the reproducibility and efficiency of experimental systems remains a critical issue. In particular, the detection of target proteins using oligo DNA-conjugated antibodies in combination with RCA often involves a trade-off between detection specificity and sensitivity. The authors aimed to address this challenge by optimizing conditions during immunostaining, specifically refining the blocking buffer, to achieve both the specific binding of oligo DNA-conjugated antibodies to targets and efficient RCA enzymatic reactions. Furthermore, they demonstrated the simultaneous detection of proteins and RNA while preserving spatial information by integrating RCA-based mRNA detection with this method.

Comment 1

While this study presents an interesting topic, many researchers have already reported simultaneous detection of proteins and RNA while maintaining spatial information. Consequently, the novelty of this work is insufficient.

Answer 1

Our method is unique as it is the only microscopy-based approach that can survey RNA and proteins with high multiplexing capacity, single-protein immunostaining and strong amplification and fast readout. All other methods cited in our manuscript lack at least one of these key features.

Liu, Y. *et al.* High-plex protein and whole transcriptome co-mapping at cellular resolution with spatial CITE-seq. *Nat. Biotechnol.* **41**, 1405–1409 (2023). Is an NGS based method with very high multiplexing but low resolution and sensitivity.

He, S. *et al.* High-plex imaging of RNA and proteins at subcellular resolution in fixed tissue by spatial molecular imaging. *Nat. Biotechnol.* **40**, 1794–1806 (2022) → Is a complex combinatorial based detection method based on conventional ISH probe (limited amplification and specificity).

In situ detection of individual mRNA molecules and protein complexes or post-translational modifications using padlock probes combined with the in situ proximity ligation assay. *Nat. Protoc.* **8**, 355–372 (2013). → The system has some similarities to Cassini, but is designed to assay proximity ligation.

Hybridization chain reaction enables a unified approach to multiplexed, quantitative, high-resolution immunohistochemistry and in situ hybridization. *Dev. Camb. Engl.* **148**, dev199847 (2021)→ This approach lacks Cassini's multiplexing capabilities.

If you believe we are missing published papers that compromise the novelty of our study, we would be happy to discuss them and explore how we might address this.

Comment 2

1. In Fig. 1a, the authors present an overview of Cassini. It would be beneficial to explicitly clarify which aspects of Cassini represent advancements over existing technologies to help readers better understand the unique features of this method.

Answer 2

We appreciate this suggestion and have revised Figure 1a to better highlight all the key features which altogether are specific to our method.

- a) (3) demonstrates that LMW is an essential component of the blocking buffer, previously unknown to provide specificity without interfering with enzymatic reactions.
- b) In (6), we show RCA amplification on conjugated antibody. To the best of our knowledge, this is the first method to show that RCA amplification can be used in fixed tissue to enable specific protein immunostaining with conjugated antibodies.

- c) (7) has been updated to show that the cycling process can be repeated as many times as needed and completed very rapidly, highlighting the multiplexing capacity of Cassini.

Comment 3

2. While numerous methods for highly multiplexed simultaneous detection of proteins and RNA have been reported, the manuscript should clearly delineate what can be uniquely achieved with Cassini compared to these existing techniques.

Answer 3

In the second paragraph of the main text, which discusses microscopy-based technologies capable of analyzing both proteins and RNA, we have added additional references and clarified the limitations of existing approaches. These technologies typically fall into three categories: (1) those that are slow with low multiplexing capacity; (2) methods that rely on protein proximity assays rather than true single-protein detection; and (3) approaches based on conventional ISH probe with limited signal amplification and/or signal specificity.

In contrast, our technology, as outlined in the abstract, is *“a new approach for straightforward, cost-effective multiplexed measurements of mRNA and protein features simultaneously. Cassini leverages rolling circle amplification (RCA), known for its robust amplification and remarkable stability even after intense stripping, to serially detect each feature in under 20 minutes”*.

Comment 4

3. The authors demonstrate the detection of up to 30 mRNAs and 2 proteins using Cassini. To better highlight the multiplexed measurement capability of this method, it would be more compelling to demonstrate its ability to detect a greater number of protein features (e.g., more than 4) compared to conventional immunostaining.

Answer 4

We appreciate this comment and agree that demonstrating protein multiplexing is important to highlight Cassini's advantages. To address this, we conducted a new experiment showcasing the simultaneous detection of five protein markers in both control and pathological human brain tissue (Supplementary Fig. 13 and 14).

Validation of the immunostaining as be done using co-immunostaining of conventional and conjugated antibodies shown in Supplementary Fig. 13

The protein multiplexed experiment is shown in Supplementary Fig. 14 and analyzed as follows:

“As expected, β-amyloid plaques were observed in Alzheimer's tissue but were absent in the control. Additionally, we confirmed proper staining of white matter (NeuN and MAP2), vasculature (Ulex), and astrocytes (GFAP) in both tissue types (Supp. Fig. 14).”

Comment 5

4. Demonstrating the capability for multi-round experiments with multi-color imaging would strengthen the manuscript. While the main text states that three channels are available per round, the multi-round images presented in Supplementary Fig. 9 only include single-color data for 10 rounds. Including multi-color images across multiple rounds would provide stronger validation of this capability.

Answer 5

We apologize for the confusion. The original Supplementary Fig. 9—which is now Supplementary Fig. 11 in our revised manuscript—shows two channels used for conventional immunostaining, while one channel is dedicated to sequential cycling

through RNA targets and conjugated antibodies. The data shown in the original Supplementary Fig. 9 (now Supplementary Fig. 11) are now stated as :

“In this configuration, one channel is dedicated to DAPI (405 nm), two channels are used for conventional immunostaining (488 nm and 561 nm), and the remaining channel (638 nm) is employed for the detection of mRNAs and conjugated antibodies.”

Figure 2 presents a distinct experiment that does not include conventional immunostaining, allowing all three channels to be used for detecting different features in each round. This is now clearly stated as:

“Without conventional immunostaining, three channels can be used to detect different features in each round. In a final experiment to demonstrate Cassini’s ease and throughput, we surveyed 30 mRNAs and 2 conjugated antibodies on a mouse brain hemisphere of ~40mm² (Fig. 2).”

Comment 6

5. There appears to be a potential inconsistency in the reported detection time. The abstract states that each feature can be detected in under 20 minutes, while the main text reports a detection rate of approximately 18 minutes per feature. However, Supplementary Protocol 37 indicates that each cycle requires 47 minutes plus 7 minutes for imaging, suggesting that the stated time may refer to per-color imaging in a three-color system. If this is the case, the total detection time per feature may be longer than implied. Clarifying this in the text would improve consistency and prevent potential misinterpretation.

Answer 6

We apologize for not stating this more clearly in our original submission. The full cycle takes 47 minutes, with an additional imaging time of all 4 colors (including DAPI) ranging from 1 to 7 minutes, resulting in a total cycle duration of approximately 48 to 54 minutes. Since we measure three features per cycle, this corresponds to a detection rate of roughly 18 minutes per feature (or <20 minutes) with long imaging time.

To make clear that the imaging time includes all 4 colors we update the text as follows:

“The full cycle time for this experiment was 47 minutes, with an imaging time across all four channels (405 nm - DAPI, 488 nm - Feature 1, 561 nm - Feature 2 and 638 nm - Feature 3) of 7 minutes, achieving a detection rate of 18 minutes per feature (Detailed workflow and protocol in supplementary material)”

Minor Comments

1. Gene names in Fig. 2 and the Supplemental Figures should be italicized. **Done**
2. In the imaging system description, “405n” should be corrected to “405 nm.” **Done**
3. More details about the lens, particularly the numerical aperture (NA), should be included. **Done**
4. The readability of the Cassini Workflow in the Supplemental Figures should be improved. **Workflow labels have been repositioned for improved visibility, and a legend has been added for clarification**

Reviewer #3 (Remarks to the Author):

Cassini: Streamlined and Scalable Method for in situ profiling of RNA and Protein

In this study, the authors develop and demonstrate a scalable method for jointly measuring RNA and proteins in intact tissue sections at a fraction of the time and cost of currently available commercial methods. By combining rolling circle amplification and an optimized immunostaining buffer, Cassini achieves rapid detection (approximately 18 minutes per feature) and maintains sensitivity comparable to established methods like HCR.

The quantitative comparison of antibody signal between conjugated vs. conventional immunostaining (Fig. 1c) is compelling.

Overall, Cassini is a useful extension to the existing suite of spatial multi-omic methods based on its streamlined protocol and significant reduction in time and cost. The manuscript is well-written and concise.

Major points:

Comment 1

A major limitation of this work is the lack of validation on non-mouse brain tissues. As this team of authors likely knows well, extending in situ methods originally developed in the context of the mouse brain does not work readily in diverse tissues. Applying Cassini to other tissues with different autofluorescence properties, which could affect sensitivity and specificity, would be more compelling to position Cassini as a generalizable method.

Answer 1

We appreciate this comment, and agree that extending Cassini beyond mouse brain is important. To address this limitation, we applied Cassini to both pathological and control human cortex samples, which are well known for their high levels of autofluorescence (Supp. Fig. 13 and 14). We assayed a total of 5 antigens, across both control and

Alzheimer's affected brain. The results highlight our ability to distinguish these five protein epitopes with different distributions within the tissue and analyzed as follows:
“As expected, β -amyloid plaques were observed in Alzheimer's tissue but were absent in the control. Additionally, we confirmed proper staining of white matter (NeuN and MAP2), vasculature (Ulex), and astrocytes (GFAP) in both tissue types (Supp. Fig. 14).”

Comment 2

Fig. 1d shows a speckled background noise in the conjugated NeuN antibody signal outside the dentate gyrus, suggesting that further refinement of the staining conditions may enhance signal specificity. A quantification of how such off-target binding confounds accurate localization of protein expression would be helpful, especially in tissues outside of the mouse brain.

Answer 2

We agree that the background signal observed in the fiber tract is unexpected for NeuN immunostaining. To address this, we have added a new supplementary figure demonstrating that the signal is not due to antibody conjugation. We now explain this in the revised manuscript as follows:

“The speckled background observed in the fiber tract appears in both conventional and conjugated NeuN immunostaining (Fig. 1d left and Supp. Fig. 6), suggesting it is not specific to the conjugated method but likely reflects non-specific binding inherent to the antibody.”

Moreover, the quantification shown in Fig. 1c demonstrates the absence of background signal in the fiber tract located dorsal to CA1.

REVIEWER COMMENTS ROUND 2

Reviewer #1 (Remarks to the Author):

My comments have been appropriately addressed in the revised version of the manuscript, and I now think it is acceptable for publication.

Reviewer #2 (Remarks to the Author):

Additional Comments

Thank you for the thorough and mostly convincing revisions. I am satisfied with the authors' responses and the new data provided for Comments 1–4 and 6, as well as the

minor points. The manuscript is now much clearer, and the added experiments substantially strengthen the paper.

Remaining issue (Comment 5 – multi-round, multi-colour imaging):

Demonstrating true multi-round and multi-color capability with raw or minimally processed images is essential for a rigorous technical evaluation and is considered standard in the spatial omics community. Currently, Supplementary Fig. 11 and Fig. 2 describe how the channels are allocated; however, they do not display representative composite images from successive rounds that show the simultaneous detection of three distinct features within each round.

The microscopy images for all genes and proteins presented in Fig. 2 are now provided in Supplementary Fig. 15.

Reviewer #3 (Remarks to the Author):

The authors have satisfactorily addressed all comments.